# Protective Effect of Low-Molecular-Weight Fucoidan on Radiation-Induced Fibrosis Through TGF-β1/Smad Pathway-Mediated Inhibition of Collagen I Accumulation

**DOI:** 10.3390/md18030136

**Published:** 2020-02-27

**Authors:** Szu-Yuan Wu, Yu-Ting Chen, Guo-Yu Tsai, Fu-Yin Hsu, Pai-An Hwang

**Affiliations:** 1Department of Food Nutrition and Health Biotechnology, College of Medical and Health Science, Asia University, Taichung 413, Taiwan; szuyuanwu5399@gmail.com; 2Division of Radiation Oncology, Lo-Hsu Medical Foundation, Lotung Poh-Ai Hospital, Yilan 265, Taiwan; 3Department of Radiation Oncology, Wan Fang Hospital, Taipei Medical University, Taipei 116, Taiwan; 4Big Data Center, Lo-Hsu Medical Foundation, Lotung Poh-Ai Hospital, Yilan 265, Taiwan; 5Department of Healthcare Administration, College of Medical and Health Science, Asia University, Taichung 413, Taiwan; 6Department of Bioscience and Biotechnology, National Taiwan Ocean University, Keelung 202, Taiwan; m3kitty41221@gmail.com (Y.-T.C.); 1073b014@mail.ntou.edu.tw (G.-Y.T.); fyhsu5565@gmail.com (F.-Y.H.); 7Center of Excellence for the Oceans, National Taiwan Ocean University, Keelung 202, Taiwan

**Keywords:** low-molecular-weight fucoidan, radiation-induced fibrosis, TGF-β1, collagen I, Smad pathway

## Abstract

Radiation-induced fibrosis (RIF) occurs after radiation therapy in normal tissues due to excessive production and deposition of extracellular matrix proteins and collagen, possibly resulting in organ function impairment. This study investigates the effects of low-molecular-weight fucoidan (LMF) on irradiated NIH3T3 cells. Specifically, we quantified cellular metabolic activity, fibrosis-related mRNA expression, transforming growth factor beta-1 (TGF-β1), and collagen-1 protein expression, and fibroblast contractility in response to LMF. LMF pre + post-treatment could more effectively increase cellular metabolic activity compared with LMF post-treatment. LMF pre + post-treatment inhibited TGF-β1 expression, which mediates negative activation of phosphorylated Smad3 (pSmad3) and Smad4 complex formation and suppresses downstream collagen I accumulation. In addition, LMF pre + post-treatment significantly reduced actin-stress fibers in irradiated NIH3T3 cells. LMF, a natural substance obtained from brown seaweed, may be a candidate agent for preventing or inhibiting RIF.

## 1. Introduction

Radiotherapy (RT) is widely used to treat various cancers; however, RT can cause varying degrees of damage to different tissues and organs, including the skin, ligaments, tendons, muscles, viscera, nerves, lungs, gastrointestinal and genitourinary tracts, and bone, depending on the treatment site [1]. Radiation-induced fibrosis (RIF) may be caused by previous acute inflammation and then result in long-term disability following RT [2]. Several cytokines, growth factors, fibroblast proliferation and differentiation, and excessive production and deposition of extracellular matrix proteins and collagen are involved in RIF [3]. Herskind et al. [4] reported that the cellular response to radiation injury results in an altered differentiation pattern of the fibroblast/fibrocyte cell system, and RT accelerates the differentiation of mitotic progenitor fibroblasts to postmitotic functional fibrocytes, resulting in the synthesis of extracellular matrix proteins, such as interstitial collagen. However, Yano et al. [5] reported another mechanism underlying the development of RIF: the mRNA level of transforming growth factor beta-1 (TGF-β1) increases earlier than that of collagen after radiation injury, and TGF-β1 is considered a master switch for the development of RIF. The complications of RIF stem from both the direct and indirect effects of progressive fibrosis on the nerves, muscles, tendons, ligaments, skin, bones, and lymphatic system. These complications of RIF are irreversible and may cause organ function impairment or organ failure, which can lead to death or significant deterioration of patients’ quality of life [6]. Due to the side effects and complications of RT, it is necessary to protect normal cells from radiation injury and further inhibit the development of RIF. 

Amifostine is a radioprotective agent approved by the U.S. Food and Drug Administration (FDA) that was reported to be effective in reducing radiation pneumonitis; however, the decrease in the incidence of fibrosis caused by amifostine did not reach statistical significance [7]. In addition, amifostine leads to toxic effects, such as nausea, vomiting, and allergy, and these side effects result in poor compliance to amifostine treatment [8]. Researchers have been showing increased interest in using natural products for treating radiation-induced injury. Studies on the isolation and production of a radioprotective agent have mainly focused on plants, such as curcumin (extracted from *Curcuma longa*) [9], quinic acid (extracted from coffee and cocoa) [10], lycopene (extracted from *Lycopersicon esculentum*) [11], rutin [12], black tea extract of *Camellia sinensis* [13], and silymarin (extracted from *Silybum marianum*) [14], and their mechanisms of action are based on antioxidant or anti-inflammatory activities. Other studies have reported that quercetin [15] and α-lipoic acid [16] prevent RIF by inhibiting TGF-β1 expression.

Fucoidan is a general term used for a class of sulfated and fucosylated polysaccharides found in brown seaweed. The biological activities of fucoidan vary with species, molecular weight, composition, structure, and extraction method [17]. The nonanimal origin of fucoidan is related to particular pharmacological activities [18]. Studies have indicated that fucoidan exerts radioprotective [19,20] and immunostimulating effects on host immune cells [21,22]. Low-molecular-weight fucoidan (LMF) was reported to reduce TGF-β1-mediated renal fibrosis in diabetic nephropathy in vitro and in vivo [23]. Furthermore, fucoidan acts as a TGF-β1 inhibitor to reduce the activation of hepatic stellate cells and the formation of liver fibrosis [24]. 

In previous studies, we demonstrated that LMF has more favorable bioactivity than high-molecular-weight fucoidan [25], with no toxicological effects found on rats after 28 days of repeated oral administration [26]. In addition, it could exert a UV-protective effect on fibroblasts [25] and prevent renal tubulointerstitial fibrosis [27]. These findings indicate that LMF may be used as an anti-RIF agent. Therefore, in the present study, we investigate the effects of LMF on irradiated NIH3T3 cells. Specifically, we quantify cellular metabolic activity, fibrosis-related mRNA expression, TGF-β1 and collagen-1 protein expression, and fibroblast contractility in response to LMF. To the best of our knowledge, no study has examined the anti-RIF effects of LMF on normal fibroblast cells.

## 2. Results and Discussion

### 2.1. LMF Reduced Cell Death in Irradiated Fibroblast Cells

Ionizing radiation is an effective DNA-damaging agent, which produces a range of lesions in cellular DNA, and DNA double-strand breaks are regarded as an important cause of cell death [28]. Therefore, a radioprotectant that effectively protects against the harmful effects of radiation is desirable. We first investigated the effects of different concentrations of LMF on cellular metabolic activity at 24 and 48 h. As shown in Figure 1, LMF did not cause toxicity in NIH3T3 cells, and cellular metabolic activity was significantly higher after treatment with 75 μg/mL of LMF. According to previous studies, fucoidan enhances the proliferation of NIH3T3 fibroblasts [29], which may be because fucoidan can significantly increase the expression of cyclin D1 and reduce the expression of p27 [30]. Our findings in proliferation of NIH3T3 fibroblasts were compatible with the two previous studies.

Kobashigawa et al. [31] pretreated and post-treated irradiated cells with ascorbic acid and reported that post-treatment with ascorbic acid was more effective in suppressing radiation-induced cellular senescence compared with pretreatment with ascorbic acid. However, pretreatment of irradiated cells with vitamin D could prevent reactive oxygen species (ROS) production, apoptosis, and senescence induced by radiation [32]. Therefore, it can be speculated that the chance of sample treatment would affect the state of radiated cells. To determine the radioprotective effect of LMF, we incubated NIH3T3 cells (pre + post-treatment) with LMF before and after irradiation (1 Gy). After 24 and 48 h, the cells were harvested and used for the cellular metabolic activity assay. The cells that were not treated with LMF (control group) showed significantly higher viability than did irradiated cells (* *p* < 0.01). In the pre + post-treatment group, LMF significantly increased the viability of irradiated cells at a dosage of 10–50 μg/mL (^#^
*p* < 0.01; Figure 2). In the radio-repair (post-treatment) assay, irradiated NIH3T3 cells were incubated with LMF for 24 and 48 h; then, cellular metabolic activity was examined. However, in the post-treatment group, the viability of irradiated cells treated with LMF did not significantly differ from that of irradiated cells not treated with LMF, except at an LMF dosage of 50 μg/mL (^#^
*p* < 0.01; Figure 3). This result indicates that LMF pre + post-treatment was more effective in increasing cellular metabolic activity compared with LMF post-treatment. Exposing HS68 human skin fibroblasts to fucoidan before irradiation could protect cells and increase their survival by two times compared with radiation alone [20]. In addition, Byon et al. [21] indicated that fucoidan exerted radioprotective effects on bone marrow cells with respect to cellular metabolic activity and immunoreactivity. Rhee and Lee [22] also reported that intraperitoneal injection of fucoidan into mice could protect against the γ-radiation-induced damage of blood cells. Therefore, fucoidan and LMF may exert a favorable radioprotective effect on cells. In further experiments, we evaluated the radioprotective effects (pre + post-treatment) of LMF on fibrosis-related mRNA expression, TGF-β1 and collagen-1 protein expression, and fibroblast contractility. 

### 2.2. LMF Inhibited Radiation-Induced Fibrosis Through the TGF-β1/Smad Signaling Pathway

TGF-β1 is a member of a superfamily of multifunctional cytokines and is a potent inducer of collagen gene expression during fibrosis, and strongly contributes to fibrosis disorders [33]. Yano et al. [5] indicated that ionizing radiation upregulated type I collagen (collagen I) expression through the TGF-β/Smad signaling pathway, and expression of TGF-β1 increased by more than 20 times that of TGF-β2. Furthermore, ERK1/2 and p38 mitogen-activated protein kinase are not involved in the development of RIF in NIH3T3 cells. Therefore, TGF-β1 is considered the main switch for the development of RIF. In the TGF-β/Smad signaling pathway, TGF-β signals from the cell surface are transduced into the nucleus by Smad proteins. Phosphorylated Smad3 (pSmad3) binds to Smad4 in the cytosol, and the pSmad3/Smad4 complex enters the nucleus to regulate collagen I expression [34]. 

First, we pre + post-treated NIH3T3 cells with LMF to determine whether LMF suppresses the TGF-β1/Smad signaling pathway, affecting radiation-induced fibrosis. As shown in Figure 4, the mRNA expression of TGF-β1, Smad3, and Smad4 was higher in the irradiation group than in the control group at 24 h and 48 h, respectively. LMF pre + post-treatment significantly reduced the TGF-β1 level at 24 h for all dosages of LMF; however, only a high dosage (50 μg/mL) of LMF reduced the TGF-β1 level 48 h after irradiation. Furthermore, LMF pre + post-treatment significantly inhibited Smad3 and Smad4 mRNA expression in a dose-dependent manner at 48 h. In most of the cases, changes in mRNA levels and protein levels do not correlate well mainly due to the regulation control at different levels [35]. Therefore, protein expression to explore the effectiveness and dosage effect of LMF on the inhibition of radiation-induced fibrosis, including the protein expression of TGF-β1, is shown in Figures 5, 7 and 8. TGF-β1/Smad signaling was demonstrated to participate in the development of RIF [5]. Thus, we determined whether LMF disrupts the formation of the pSmad3/Smad4 complex and its entry into the nucleus. In the immunofluorescence experiment, the expression of pSmad3 and Smad4 protein was higher in the irradiation group than in the control group (Figure 5A,B, * *p* < 0.01), and Smad4 was clearly observed in the nucleus (Figure 5C, * *p* < 0.01). LMF pre + post-treatment at a moderate and high dosage significantly reduced the expression of pSmad3 in cells (Figure 5B, ^#^
*p* < 0.01) and inhibited the migration of Smad4 into the nucleus (Figure 5C, ^#^
*p* < 0.01). This finding is consistent with those reported by Chen et al. [23,27]; they indicated that administering LMF to mice or rats with chronic kidney disease could reduce TGF-β1 expression and inhibit the Smad signaling pathway, preventing tubulointerstitial fibrosis. Kim et al. [36] also reported that fucoidan binds to TGF-β1, but not to the TGF-β receptor, to inhibit the activation of the downstream signaling pathway. Thus, LMF pre + post-treatment not only reduced TGF-β1 mRNA expression in irradiated NIH3T3 cells but also inhibited the interaction of TGF-β1 with its receptor, blocking the TGF-β1/Smad signaling pathway.

### 2.3. LMF Inhibited TGF-β1 Protein Activation and *Extracellular Matrix Formation* in Irradiated Fibroblast Cells

TGF-β is responsible for many functions of RIF pathogenesis, including the differentiation of fibroblasts into myofibroblasts [37], wherein phenotypic changes in fibroblasts result in an increase in α-smooth muscle actin (α-SMA) expression. α-SMA subsequently converts primitive fibroblasts into matured myofibroblasts [38]. These myofibroblasts can also be derived from fibroblasts or epithelial cells undergoing epithelial–mesenchymal transition (EMT) [39]. In response to TGF-β, myofibroblasts secrete extracellular matrix proteins, including collagen, fibronectin, and proteoglycan, and by doing so is in charge of fibrosis [40].

Next, we pre + post-treated NIH3T3 cells with LMF to determine whether LMF inhibits TGF-β1 protein activation and reduces extracellular matrix formation in irradiated cells. To our knowledge, the effects of fucoidan on radiation-induced extracellular matrix formation have not been determined. As shown in Figure 6, collagen I, fibronectin, and α-SMA mRNA expression were higher in the irradiation group than in the control group, and the mRNA expression of collagen I, fibronectin, and α-SMA increased at 24 h and continued to increase at 48 h (* *p* < 0.01). LMF pre + post-treatment significantly reduced the mRNA expression of collagen I, fibronectin, and α-SMA at 24 and 48 h (* *p* < 0.01). Furthermore, TGF-β1 mRNA expression significantly reduced at 24 h and collagen I mRNA expression markedly reduced at 48 h in the LMF pre + post-treatment group. Thus, we then determined TGF-β1 and collagen I protein expression through Western blotting at 48 h to confirm the effects of LMF on the progression of RIF from upstream (TGF-β1 protein expression) to downstream (collagen I protein expression). When irradiated cells were pre + post-treated with 25 μg/mL LMF could most effectively suppress TGF-β1 and collagen I protein expression. On the other hand, the effectiveness of TGF-β1 and collagen I at 50 μg/mL LMF decreased (Figure 7). In our previous study on tubulointerstitial fibrosis, we demonstrated that LMF could reduce TGF-β, collagen I, fibronectin, and α-SMA protein expression in vivo [27]. Chen et al. [23] reported that LMF effectively reduced TGF-β protein expression and extracellular matrix accumulation in fibrogenesis in the kidneys of type 2 diabetic rats. In addition, we also found that low-dose treatment of LMF significantly reduced TGF-β1 expression and promoted renal function in CKD mice, but high-dose treatment of LMF only moderately reduced TGF-β 1 expression and did not improve renal function [27]. Therefore, our findings were compatible with previous studies and we suggest that the dosage should be carefully considered in the application of LMF to patients with radiation-induced fibrosis.

### 2.4. LMF Reduced the Contractility of Irradiated Fibroblast Cells

Ionizing radiation was reported to induce fibroblast-to-myofibroblast differentiation, collagen production, and tissue contraction during radiation damage both in vitro [41] and in vivo [15]. During normal wound healing, myofibroblasts produce extracellular matrix proteins that are essential for wound contraction. However, during the pathological process of fibrosis, myofibroblasts persist and contribute to prolonged matrix protein generation, deposition, and accumulation [42], eventually leading to the loss of normal tissue architecture and function [6]. Therefore, in the present study, we determined the inhibitory effect of LMF on the differentiation and contraction process of irradiated fibroblast cells.

We found that collagen I protein expression was higher in the irradiation group than in the control group (Figure 8A,B, * *p* < 0.01) and that LMF pre + post-treatment significantly reduced collagen I expression in cells (Figure 8B, ^#^
*p* < 0.01). These data are consistent with those presented in Figure 6 and Figure 7. Furthermore, the morphology of cells differed between the LMF pre + post-treatment group and the irradiation group. In the immunofluorescence experiment of the cell skeleton (F-actin), normal NIH3T3 cells spread out and developed a spindle-like morphology with long cell arms. However, cells irradiated with 1 Gy for 48 h were small, contracted, and tended to be round, and the ratio of long axis to short axis of cells was low. Horton et al. [15] explained that the arrays of actin microfilaments are affected by irradiation, and cells have prominent actin-stress fibers and filopodial extensions. LMF pre + post-treatment significantly reduced prominent actin-stress fibers and filopodial extensions in irradiated cells (Figure 8A), and the ratio of long axis to short axis of cells became higher (Figure 8C). Therefore, ROS also play an important role in TGF-β1-induced EMT primarily through activation of MAPK and subsequently through ERK-directed activation of Smad pathway in proximal tubular epithelial cells [43]. TGF-β1 also plays a central role in the activation of p53 and ataxia-telangiectasia-mutated (ATM) signaling pathways [44]. In the previous study, LMF promoted p53 accumulation, p21 expression, and significant decreases in ataxia-telangiectasia-mutated (ATM), checkpoint kinase 1 (Chk1), and γ-H2AX phosphorylation in p53+/+ cells compared with p53−/− cells [45]. LMF also reduced the expression of TGF-β1 in NRK-52E cells and CKD mice [27]. In addition, LMS is also a potential drug to decrease the formation of ROS, and augment cell survival and proliferation [46]. In the aforementioned studies, LMS’ regulation of the expression of p53 and ROS was established well [11,12]. Thus, we did not repeat similar experiments but focused on radiation-induced fibrosis through TGF-β1/Smad pathway. These findings suggest that LMF can suppress or reverse fibrosis in irradiated cells by inhibiting fibroblast contractility and collagen I accumulation. 

## 3. Materials and Methods

### 3.1. LMF Preparation

LMF extracted from *Laminaria japonica* seaweed sample was ground to powder using a mini blender and then dried using a dryer at 50 °C. Then, 100 g of dried seaweed were treated with 5 L of distilled water and boiled at 100 °C for 30 min; the extract was centrifuged at 10,000× *g* for 20 min. The supernatant was incubated with 4 M CaCl_2_ for 1 h to separate alginic acid and recentrifuged at 10,000× *g* for 20 min. All polysaccharides were dialyzed (a cutoff of 10,000 Da) using deionized water for 48 h, precipitated by adding ethanol at a ratio of 1:3 (V/V), and then left overnight to obtain crude fucoidan. The fucoidan that was obtained was fractionated through anion-exchange chromatography performed using DEAE-Sephadex A-25 (Cl^−^ form, Pharmacia, Uppsala, Sweden) through elution with 1.5 M NaCl. The fraction was collected and hydrolyzed with a glycolytic enzyme to obtain an average molecular weight of <3000 Da, resulting in an LMF sample. The LMF sample had an average molecular weight of 1.2 kDa (90.1%) [25], a fucose content of 396.2 ± 10.6 μmol/g [47], and a sulfate content of 35.4% ± 1.3% (w/w) [48]. The LMF sample was completely dissolved in phosphate-buffered saline (PBS) by stirring at room temperature for 30 min.

### 3.2. Cell Culture

NIH3T3 cells, a primary mouse fibroblast cell line, were obtained from the Bioresource Collection and Research Center (60008, BCRC, Hsinchu, Taiwan) and cultured in DMEM supplemented with 10% fetal bovine serum (FBS), 1.5 g/L of NaHCO_3_, 1% GlutaMAX, and 1% penicillin-streptomycin at 37 °C in an atmosphere of 5% CO_2_. The cells were subcultured following trypsinization, and the medium was renewed every 2 days.

### 3.3. Cellular Metabolic Activity after LMF Treatment

NIH3T3 cells were seeded into 96-well plates at a density of 4 × 10^4^ cells/well, cultured for 24 h, and then incubated for 24 and 48 h with or without different concentrations (10, 25, and 50 μg/mL) of LMF. At the required time point, cells were reacted with MTT (1 mg/mL) for 4 h, and absorbance was recorded at 570 nm [49]. Cellular metabolic activity (%) was determined as (A1/A0) × 100%, where A0 and A1 were the absorbance of the control group (absence of LMF) and other groups, respectively.

### 3.4. Radioprotection (Pre + Post-Treatment) and Radio-repair (Post-Treatment) Assay

NIH3T3 cells were seeded into 96-well plates at a density of 4 × 10^4^ cells/well and cultured for 24 h. The experiments were categorized into two groups: pre + post-treatment and post-treatment. In the pre + post-treatment group experiments, the radioprotective effect of LMF was evaluated. NIH3T3 cells were incubated with LMF for 24 h. Before and after 1 Gy irradiation within 1 s [50], the medium was substituted with fresh medium. After 1 Gy irradiation, the medium was removed and NIH3T3 cells were incubated with LMF for 24 and 48 h to perform the cellular metabolic activity assay. In the post-treatment group experiments, the radio-repair effect of LMF was evaluated. Before and after 1 Gy irradiation, the medium was substituted with fresh medium. After 1 Gy irradiation, the medium was removed and NIH3T3 cells were incubated with LMF for 24 and 48 h to perform the cellular metabolic activity assay. For irradiation, cells were exposed to 1 Gy gamma rays produced from Elekta SLi-25 Linear Accelerator (Stockholm, Sweden).

### 3.5. Fibrosis-related mRNA expression 

NIH3T3 (4 × 10^4^ cells/well) cells were inoculated with an LMF pretreatment model. Total RNA was isolated using RNAzol B (Amersham Pharmacia Biotech, Uppsala, Sweden), and the total RNA concentration was detected using a spectrophotometer (Hitachi, Japan). cDNA was synthesized using Improm-II TM Reverse Transcriptase (Promega, Madison, WI, USA) according to the manufacturer’s instructions. PCR was performed on the reverse-transcribed cDNA product by using a thermal cycler (Biometra, UNO-Thermoboblock, Glasgow, UK) to determine the expression of TGF-β1, Smad3, Smad4, collagen I, fibronectin, α-SMA, and glyceraldehyde-3-phosphate dehydrogenase (GAPDH; as an internal control). Primer sequences used to amplify the desired cDNA are listed in Table 1. These primers were purchased from Mission Biotech Co. Ltd. (Taipei, Taiwan). PCR products were separated through electrophoresis on 1.2% agarose gel and visualized using ethidium bromide staining under UV irradiation. The image of the resulting gel was captured and analyzed using ImageMaster VDS and ImageMaster 1D Elite software (Amersham Pharmacia Biotech, Uppsala, Sweden).

### 3.6. Western Blot Assay

NIH3T3 (4 × 10^4^ cells/well) cells were inoculated with an LMF pretreatment model for 48 h. Cellular proteins were extracted using a lysis buffer (10 mM Tris-HCl (pH 7.8), 5 mM MgCl_2_, 0.3 mM ethylene glycol tetraacetic acid, 10 mM KCl, 1 mM dithiothreitol, 0.1 mM phenylmethylsulfonyl fluoride, and protease inhibitors). The protein content was determined using the Bio-Rad DC (detergent-compatible) Protein Assay (Bio-Rad, Berkely, CA, USA). Fifty micrograms of cellular protein were separated on an 8%–15% SDS-polyacrylamide gel and transferred onto a polyvinylidene difluoride (PVDF) membrane (Bio-Rad, Berkely, CA, USA). The immunoblot was incubated overnight with a blocking solution (5% skimmed milk at pH 7.5, 20 mM Tris, 150 mM NaCl, 0.2% Tween 20), followed by incubation with a primary antibody at 4 °C overnight. A secondary antibody was incubated with the PVDF membrane at room temperature for 1 h. Protein expression was detected through staining with nitro blue tetrazolium (NBT)/ 5-bromo-4-chloro-3-indolyl phosphate (BCIP) (Pierce, Waltham, MA, USA) and quantitated by performing a densitometric analysis (Pharmacia, Imagemaster VDS, Uppsala, Sweden). Primary antibodies, namely the TGF-β1 antibody (sc-52893, Santa Cruz Biotechnology, Santa Cruz, CA, USA), collagen I antibody (sc-59772, Santa Cruz Biotechnology, Santa Cruz, CA, USA), and GAPDH antibody (Santa Cruz Biotechnology), were applied at a 1:1000 dilution. Secondary antibody, namely antimouse IgG-peroxidase (from goat; Sigma, St. Louis, MO, USA), was applied at a 1:5000 dilution. The images of resulting gels were captured and analyzed using ImageMaster VDS and ImageMaster 1D Elite software (Amersham Pharmacia Biotech, Uppsala, Sweden).

### 3.7. Immunocytochemistry

NIH3T3 (4 × 10^4^ cells/well) cells were plated on collagen-coated glass chamber slides (Corning^®^ BioCoat™, Corning, NY, USA) and then inoculated with an LMF pretreatment model for 48 h. The cells were washed with ice-cold PBS, fixed for 10 min in 3.6% formaldehyde in PBS, quenched for 10 min with 0.3 M glycine in PBS twice, and permeabilized with 0.1% Triton-X 100 for 5 min. The cells were blocked with 1% BSA in PBS for 1 h at room temperature and then incubated with a 1:500 dilution of primary antibodies, namely pSmad3 antibody (Bioss International, London, UK), Smad4 antibody (sc-7966, Santa Cruz Biotechnology), and collagen I antibody (sc-59772, Santa Cruz Biotechnology) for 1 h. Immunoreactivity was visualized by incubating the cells with an FITC (AP132F, Merck, Darmstadt, DE, USA) or Alexa Fluor^TM^ 594 (Thermo Scientific, Waltham, MA, USA) conjugated secondary antibody diluted in a blocking buffer for 1 h. The F-actin cytoskeleton was stained through incubation with Alexa Fluor^TM^ 488 phalloidin (Thermo Scientific, Waltham, MA, USA) for 30 min and 50 nM TRIT-C phalloidin (Sigma) for 15 min. Cell nuclei were stained with 10 ng/mL of DAPI (Sigma) for 30 min in PBS and cover slipped with 50% glycerol. All images were captured using a confocal laser scanning microscope (Leica TCS SP8, Wetzlar, Germany) at 630× magnification and merged with Image J software for the qualitative assessment of cytoskeletal morphology, the ratio of long axis to short axis of cells, and the localization of pSmad3, Smad4, and collagen I.

### 3.8. Statistical Analyses

Numerical data are presented as the mean ± standard deviation. Data are representative of three independent experiments, and statistical analysis was performed using Student’s t test. A probability level of *p* < 0.01 was considered significant.

## 4. Conclusions

In this study, we investigated the effects of LMF on cellular metabolic activity, fibrosis-related mRNA expression, TGF-β1 and collagen-1 protein expression, and fibroblast contractility in irradiated NIH3T3 cells. Our results showed that LMF pre + post-treatment could more effectively increase cellular metabolic activity compared with LMF post-treatment. LMF pre + post-treatment inhibited TGF-β1 expression, which mediated negative activation of pSmad3 and Smad4 complex formation and suppressed downstream collagen I accumulation. In addition, LMF pre + post-treatment significantly reduced actin-stress fibers in irradiated cells. LMF, a natural substance obtained from brown seaweed, may be a candidate agent for preventing or inhibiting the progression of RIF.

## Figures and Tables

**Figure 1 marinedrugs-18-00136-f001:**
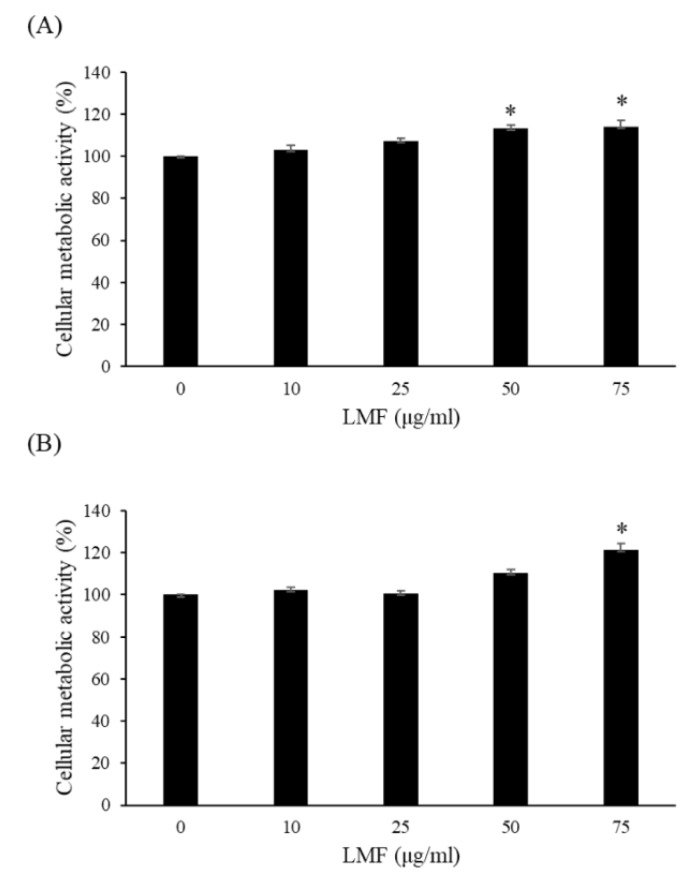
Low-molecular-weight fucoidan (LMF) caused no toxicity in NIH3T3 cells after (**A**) 24 h and (**B**) 48 h of treatment. NIH3T3 cells were incubated for 24 and 48 h with various concentrations of LMF (0, 10, 25, 50, and 75 μg/mL), and cellular metabolic activity was measured using the MTT assay. Values are expressed as the mean ± standard error of three independent experiments. * *p* < 0.01 compared with the control (0 μg/mL of LMF, 0 Gy).

**Figure 2 marinedrugs-18-00136-f002:**
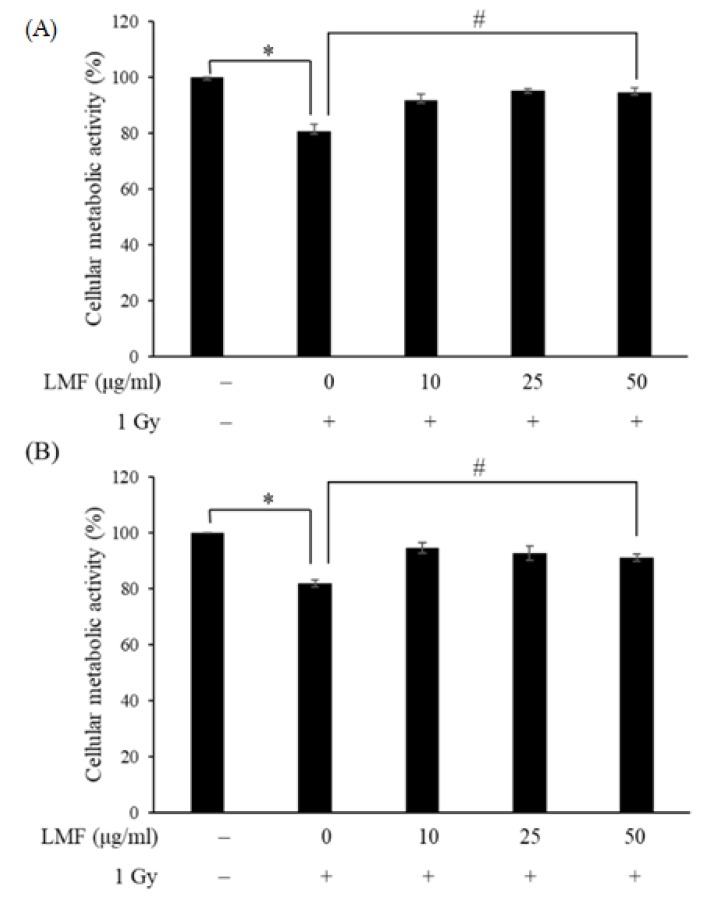
Effects of LMF pre + post-treatment on cellular metabolic activity (**A**) 24 h and (**B**) 48 h after γ-irradiation. NIH3T3 cells were incubated with LMF for 24 h. Before and after 1 Gy irradiation, the medium was substituted with fresh medium. After 1 Gy irradiation, the medium was removed and NIH3T3 cells were incubated with LMF for 24 and 48 h. Then, cellular metabolic activity was examined using the MTT assay. Values are expressed as the mean ± standard error of three independent experiments. * *p* < 0.01 compared with the control (0 μg/mL of LMF, 0 Gy), ^#^
*p* < 0.01 compared with the irradiation control (0 μg/mL of LMF, 1 Gy).

**Figure 3 marinedrugs-18-00136-f003:**
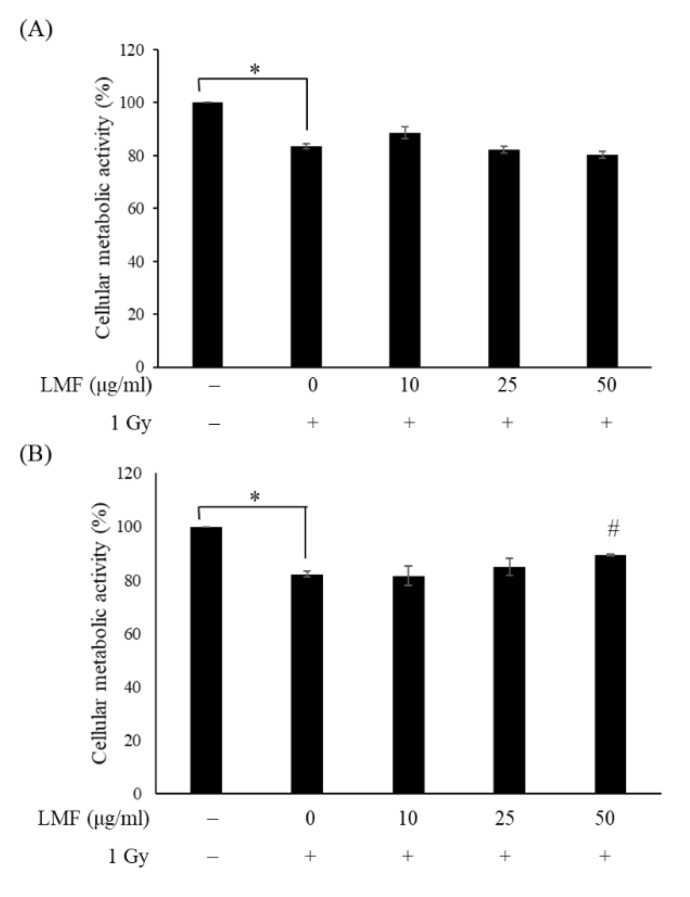
Effects of LMF post-treatment on cellular metabolic activity (**A**) 24 h and (**B**) 48 h after γ-irradiation. After 1 Gy irradiation, the medium was removed and NIH3T3 cells were incubated with LMF for 24 and 48 h. Cellular metabolic activity was measured using the MTT assay. Values are expressed as the mean ± standard error of three independent experiments. * *p* < 0.01 compared with the control (0 μg/mL of LMF, 0 Gy), ^#^
*p* < 0.01 compared with the irradiation control (0 μg/mL of LMF, 1 Gy).

**Figure 4 marinedrugs-18-00136-f004:**
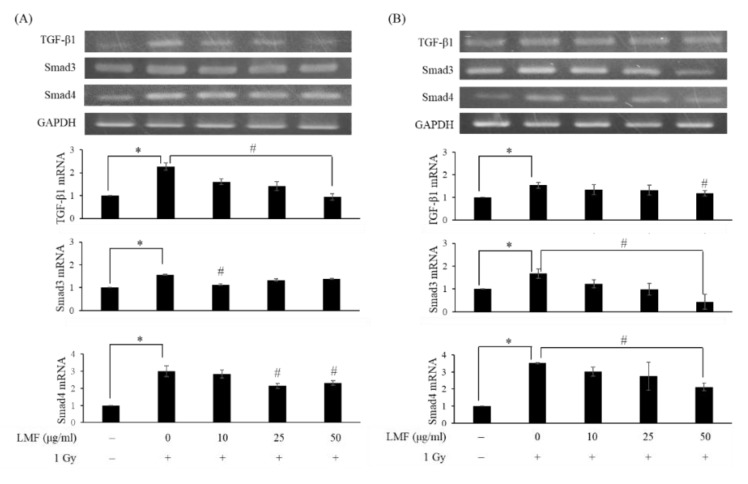
LMF pre + post-treatment represses the activation of transforming growth factor beta-1 (TGF-β1) and the pro-fibrosis markers Smad3 and Smad4 mRNA expression and its relative fold (**A**) 24 h and (**B**) 48 h after γ-irradiation. NIH3T3 cells were incubated with LMF for 24 h before 1 Gy irradiation. After irradiation, the medium was substituted with fresh medium, and cells were then incubated with LMF for 24 and 48 h. Total RNA was isolated and reverse transcribed. The resulting cDNA was then subjected to PCR with the indicated primers. Values are expressed as the mean ± standard error of three independent experiments. * *p* < 0.01 compared with the control (0 μg/mL of LMF, 0 Gy), ^#^
*p* < 0.01 compared with the irradiation control (0 μg/mL of LMF, 1 Gy).

**Figure 5 marinedrugs-18-00136-f005:**
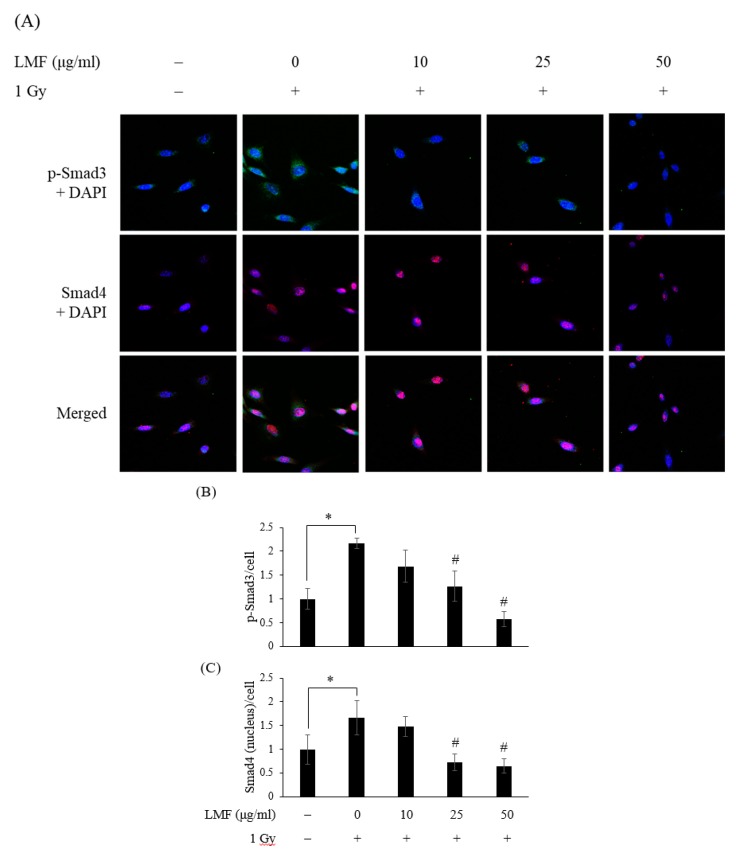
LMF pre + post-treatment represses the protein expression of pSmad3 and Smad4 48 h after γ-irradiation. NIH3T3 cells were incubated with LMF for 24 h before 1 Gy irradiation. (**A**) Immunocytochemistry demonstrated that LMF reduced pSmad3 formation and inhibited the migration of Smad4 into the nucleus. Representative images show localization of nuclei (blue = DAPI), pSmad3 (green = FITC conjugate), and Smad4 (red = Alexa Fluor^TM^ 594) at 630× magnification. Graphs quantifying the relative fold of (**B**) pSmad3 and (**C**) Smad4 within the nucleus. Values are expressed as the mean ± standard error of three independent experiments. * *p* < 0.01 compared with the control (0 μg/mL of LMF, 0 Gy), # *p* < 0.01 compared with the irradiation control (0 μg/mL of LMF, 1 Gy).

**Figure 6 marinedrugs-18-00136-f006:**
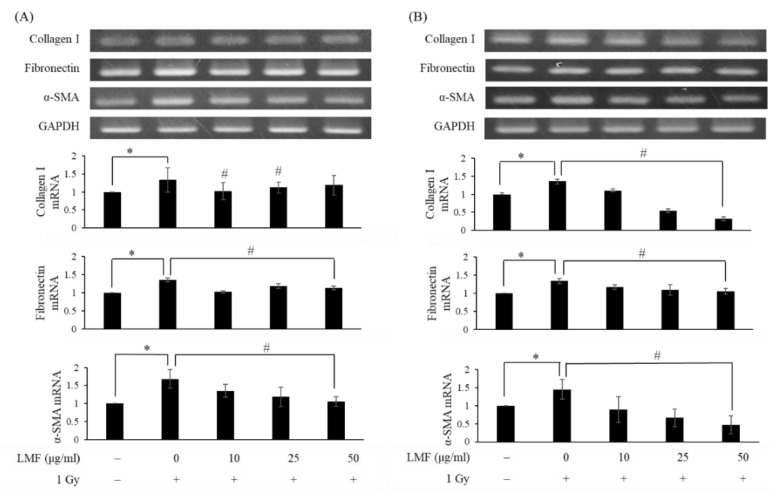
LMF pre + post-treatment represses the mRNA expression of the fibrosis markers collagen I, fibronectin, and α-smooth muscle actin (α-SMA) and its relative fold (**A**) 24 h and (**B**) 48 h after γ-irradiation. NIH3T3 cells were incubated with LMF for 24 h before 1 Gy irradiation. After irradiation, the medium was substituted with fresh medium, and cells were then incubated for 24 and 48 h. Total RNA was isolated and reverse transcribed. The resulting cDNA was then subjected to PCR with the indicated primers. Values are expressed as the mean ± standard error of three independent experiments. * *p* < 0.01 compared with the control (0 μg/mL of LMF, 0 Gy), ^#^
*p* < 0.01 compared with the irradiation control (0 μg/mL of LMF, 1 Gy).

**Figure 7 marinedrugs-18-00136-f007:**
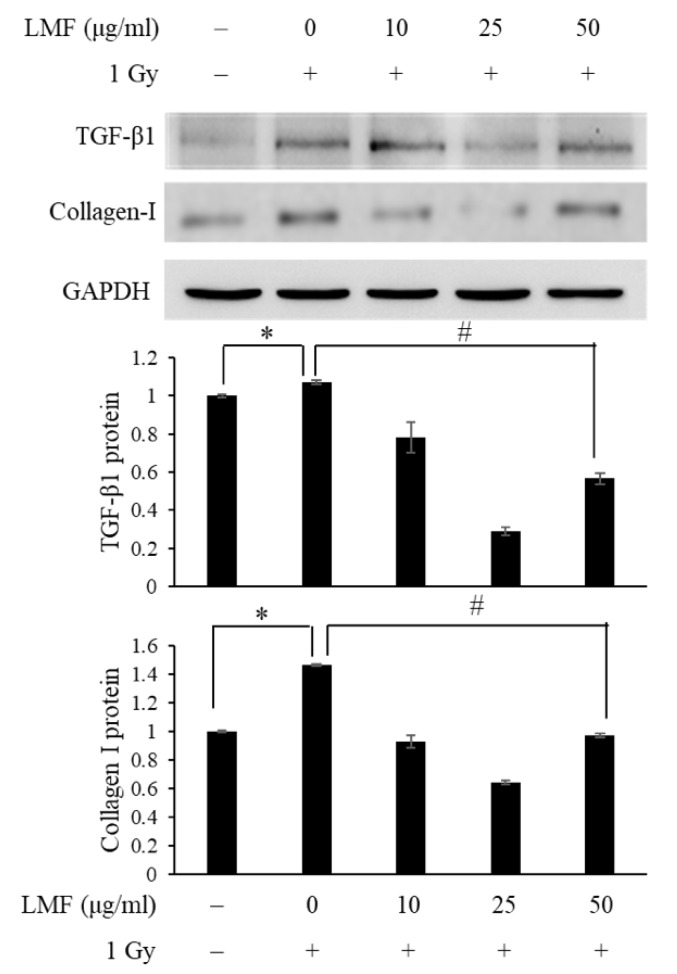
LMF pre + post-treatment represses the protein expression of TGF-β1 and collagen I and its relative fold after γ-irradiation for 48 h. NIH3T3 cells were incubated with LMF for 24 h before 1 Gy irradiation. Total cell lysates were subjected to SDS-PAGE, transferred, and probed with the indicated antibodies. Glyceraldehyde-3-phosphate dehydrogenase (GAPDH) was used as an internal control for the Western blot assay. Then, 15% polyacrylamide gels were transferred onto membranes and probed with anti-TGF-β1 and anti-collagen I antibodies. Values are expressed as the mean ± standard error of three independent experiments. * *p* < 0.01 compared with the control (0 μg/mL of LMF, 0 Gy), ^#^
*p* < 0.01 compared with the irradiation control (0 μg/mL of LMF, 1 Gy).

**Figure 8 marinedrugs-18-00136-f008:**
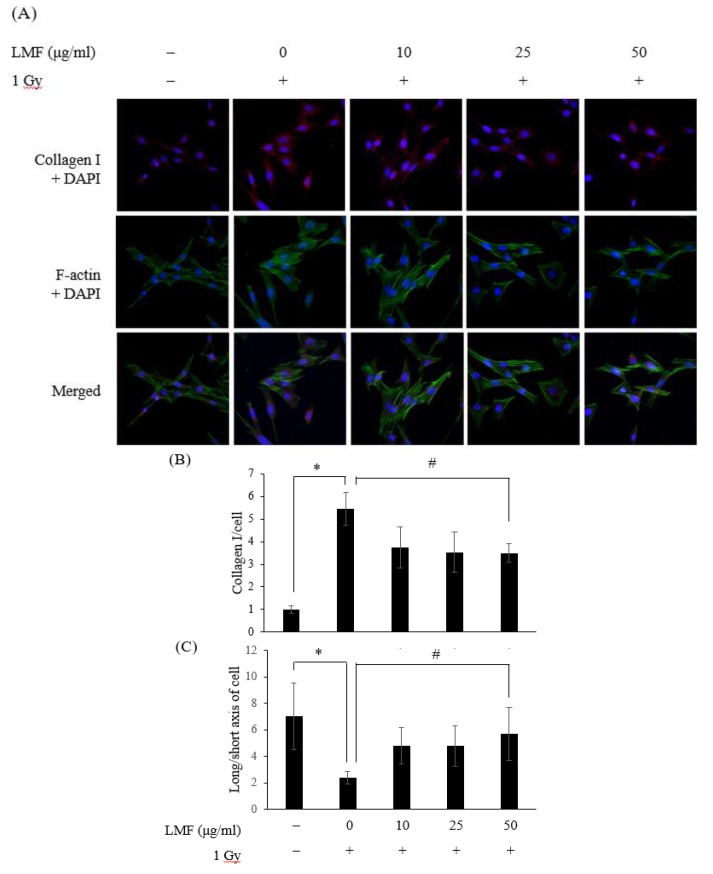
LMF pre + post-treatment represses the protein expression of collagen I and cell contractility 48 h after γ-irradiation. NIH3T3 cells were incubated with LMF for 24 h before 1 Gy irradiation. (**A**) Immunocytochemistry demonstrated that LMF reduced collagen I formation and inhibited cell contractility. Representative images show localization of nuclei (blue = DAPI), collagen I (red = Alexa Fluor^TM^ 594), and F-actin (green = Alexa Fluor^TM^ 488 phalloidin) at 630× magnification. (**B**) Graphs quantifying the relative fold of collagen I and the ratio of long axis to short axis of cells. Values are expressed as the mean ± standard error of three independent experiments. * *p* < 0.01 compared with the control (0 μg/mL of LMF, 0 Gy), ^#^
*p* < 0.01 compared with the irradiation control (0 μg/mL of LMF, 1 Gy).

**Table 1 marinedrugs-18-00136-t001:** Oligonucleotide sequences of fibrosis makers.

Gene		Primer sequence (5′- 3′)	Annealing	
TGF-β1	Sense Antisense	AGGGGCCTCTAAGAGCAGTC AGGTTGGCATTCCACTTCAC	94 °C, 30 s 55 °C, 30 s 72 °C, 30 s	25 cycle
Smad3	Sense Antisense	GGGCCAACAAGTCAACAAGT CTGGCTGGCTAAGGAGTGAC	94 °C, 30 s 55 °C, 30 s 72 °C, 30 s	25 cycle
Smad4	Sense Antisense	CGGCCGTGGCAGGGAACA CTGCAGAGCTCGGTGAAGGTGAAT	94 °C, 30 s 55 °C, 30 s 72 °C, 30 s	25 cycle
Collagen-I	Sense Antisense	CCGTGCTTCTCAGAACATCA CTTGCCCCATTCATTTGTCT	94 °C, 30 s 55 °C, 30 s 72 °C, 30 s	25 cycle
Fibronectin	Sense Antisense	GTGGCTGCCTTCAACTTCTC TGAATGCCAGTCCTTTAGGG	94 °C, 30 s 53 °C, 30 s 72 °C, 30 s	25 cycle
α-SMA	Sense Antisense	ACTGGGACGACATGGAAAAG CATCTCCAGAGTCCAGCACA	94 °C, 30 s 53 °C, 30 s 72 °C, 30 s	25 cycle
GAPDH	Sense Antisense	TGTTCCTACCCCCAATGTGT CCCTGTTGCTGTAGCCGTAT	94 °C, 30 s 55 °C, 30 s 72 °C, 30 s	25 cycle

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
