# Peer review of "Protective Effect of Low-Molecular-Weight Fucoidan on Radiation-Induced Fibrosis Through TGF-β1/Smad Pathway-Mediated Inhibition of Collagen I Accumulation"

_marinedrugs, 2020, doi:10.3390/md18030136_

Round 1
Reviewer 1 Report
In this work the authors describe the effect of a brown seaweed derived polysaccharide (fucoidan) on irradiated cells. Results show that the pre and post treatment of the cells with fucoidan inhibits effectively the production of mediators involved in the progression of the radiation induced fibrosis. The study sounds scientifically, the manuscript is well written and the experiments are well described.
However the reviewer would like to ask for some clarifications in order to improve the quality of the paper:
Line 83: is there any reason why the presence of fucoidan increases the viability of the non-irradiated cells over time?
Line 141: The authors stated that the mRNA expression of Smad3 and Smad4 increased after 24h and even more after 48h when the cells are irradiated but not treated with fucoidan. However fig.4 shows no difference between 24h and 48h of incubation period. Is there any statistical study for comparing those values?
Line 142: do the authors have any explanation about the reduction of the effectiveness of the LMF in the TGF-beta1 mRNA expression after 48h (except for the highest does)?
Figure 7: Why does protein expression firstly decrease when a LMF concentration of 25 ug/ml is used and increase again at higher concentration?
Line 293: how long have the cells been irradiated?
Author Response
Reviewer 1
Comments and Suggestions for Authors
In this work the authors describe the effect of a brown seaweed derived polysaccharide (fucoidan) on irradiated cells. Results show that the pre and post treatment of the cells with fucoidan inhibits effectively the production of mediators involved in the progression of the radiation induced fibrosis. The study sounds scientifically, the manuscript is well written and the experiments are well described.
However the reviewer would like to ask for some clarifications in order to improve the quality of the paper:
Response:
Thank you for your appreciate for our study and we will try our best to do further investigations in the field of radiation fibrosis to improve radiation related toxicity in irradiated patients in the future.
- Line 83: is there any reason why the presence of fucoidan increases the viability of the non-irradiated cells over time?
Response: Thank you very much
- According to the previous studies, fucoidan can enhance the proliferation of NIH3T3 fibroblasts [29], which may be because fucoidan can significantly increase the expression of cyclin D1 and reduce the expression of p27 [30]. Our findings in proliferation of NIH3T3 fibroblasts were compatible with the two previous studies. And we added the above sentence to the manuscript in Line 87-91 with red mark.
- Reference:
- Goonoo, N.; Bhaw-Luximon, A.; Jonas, U.; Jhurry, D.; Schönherr, H. Enhanced differentiation of human preosteoblasts on electrospun blend fiber mats of polydioxanone and anionic sulfated polysaccharides. ACS Biomater. Sci. Eng. 2017, 3, 3447-3458.
- Song, Y.S.; Li, H.; Balcos, M.C.; Yun, H.Y.; Baek, K.J.; Kwon, N.S.; Choi, H.R.; Park, K.C.; Kim, D.S. Pharmacology. Fucoidan promotes the reconstruction of skin equivalents. Korean J. Physiol. Pharmacol. 2014, 18, 327-331.
- Line 141: The authors stated that the mRNA expression of Smad3 and Smad4 increased after 24h and even more after 48h when the cells are irradiated but not treated with fucoidan. However fig.4 shows no difference between 24h and 48h of incubation period. Is there any statistical study for comparing those values?
Response: Thank you
The generation of cells at 24 h and 48 h were not the same, thus we did not compare the data for these two time points. We are sorry that he original meaning was unclear, and we have revised our sentences in Line 155-157 with red mark as followings:“As shown in Fig. 4, the mRNA expression of TGF-β1, Smad3, and Smad4 was higher in the irradiation group than in the control group in 24h and 48 h, respectively”
- Line 142: do the authors have any explanation about the reduction of the effectiveness of the LMF in the TGF-beta1 mRNA expression after 48h (except for the highest does)?
Response: Thank you.
- A key assumption in studying mRNA expression is that it is informative in the prediction of protein expression. However, only limited studies have explored the mRNA-protein expression correlation in yeast or human tissues and the results have been relatively inconsistent. In most of the cases, change in mRNA levels and protein levels do not correlate that well mainly due to the regulation control at different levels (Guo et al., 2008). Therefore, the protein expression to explore the effectiveness and dosage effect of LMF on the inhibition of radiation-induce fibrosis including the protein expression of TGF-β1 in Fig. 5, 7, 8.
- We added more discussion in Line 160-163 to the manuscript as followings: “In most of the cases, change in mRNA levels and protein levels do not correlate that well mainly due to the regulation control at different levels [35]. Therefore, the protein expression to explore the effectiveness and dosage effect of LMF on the inhibition of radiation-induce fibrosis including the protein expression of TGF-β1 in Fig. 5, 7, 8.”
- Reference:
- Guo, Y.; Xiao, P.; Lei, S.; Deng, F.; Xiao, G.G.; Liu, Y.; Chen, X.; Li, L.; Wu, S.; Chen, Y.; Jiang, H.; Tan, L.; Xie, J.; Zhu, X.; Liang, S.; Deng, H. How is mRNA expression predictive for protein expression? A correlation study on human circulating monocytes. Acta Biochim. Biophys. Sin. 2008, 40, 426-436.
- Figure 7: Why does protein expression firstly decrease when a LMF concentration of 25 ug/ml is used and increase again at higher concentration?
Response: Thank you.
- In our previous study on tubulointerstitial fibrosis (reference 27) was found that low dose treatment of LMF significantly reduced TGF-β1 expression and promoted renal function in CKD mice, but high dose LMF only moderately reduced TGF-β 1 expression and did not improve renal function. Therefore, the dosage should be carefully considered in the application of LMF to patients with radiation-induce fibrosis. And in cell culture studies, fucoidan doses of 100 μg/mL or higher are often applied to studies of diseases such as anti-tumor (Yan et al., 2019), anti-inflammation (Hwang et al., 2017) and osteoporosis (Hwang et al., 2016), and exerts significant protective effects.
Yan, M. D., Lin, H. Y., Hwang, P. A. The anti-tumor activity of brown seaweed oligo-fucoidan via lncRNA expression modulation in HepG2 cells. Cytotechnology. DOI: 10.1007/s10616-019-00293-7. 2019.
Hwang, P. A., Yan, M. D., Kuo, K. L., Phan, N. N., Lin, Y. C. A mechanism of low molecular weight fucoidans degraded by enzymatic and acidic hydrolysis for the prevention of UVB damage. J. Appl. Phycol. 29: 521-529. 2017.
Hwang, P. A., Phan, N. N., Lu, W. J., Hieu, B. T. N., Lin, Y. C. Low-molecular-weight fucoidan and high-stability fucoxanthin from brown seaweed exert prebiotics and anti-inflammatory activities in Caco-2 cells. Food & Nutrition Research. 60: 32033.
- We have revised the sentence in the section of discussion (Line 220-230 with red mark) as followings: “When irradiated cells were pre+post-treated with 25 μg/mL LMF could most effectively suppress TGF-β1 and collagen I protein expression. On the other hand, the reduction of the effectiveness of TGF-β1 and collagen I at 50 μg/mL LMF was decreased (Fig. 7). In our previous study on tubulointerstitial fibrosis, we demonstrated that LMF could reduce TGF-β, collagen I, fibronectin, and α-SMA protein expression in vivo [27]. Chen et al. [23] also reported that LMF effectively reduced TGF-β protein expression and extracellular matrix accumulation in fibrogenesis in the kidneys of type 2 diabetic rats. In addition, we also found that low dose treatment of LMF significantly reduced TGF-β1 expression and promoted renal function in CKD mice, but high dose LMF only moderately reduced TGF-β 1 expression and did not improve renal function [27]. Therefore, our findings were compatible with the previous studies and we suggest the dosage should be carefully considered in the application of LMF to patients with radiation-induce fibrosis.”
- Line 293: how long have the cells been irradiated?
Response: Thank you
- According to our irradiation protocol in previous in vitro study (Yu et al., 2018), all cells in the study were irradiated in irradiation dose of 1 Gy using a linear accelerator (Elekta, 6-MV photon beam, Crawley, UK) within one second. We revised the sentence in Line 324, and added the reference (50).
- Reference:
- Yu, H.H.; Chengchuan, K.O.; Chang, C.L.; Yuan, K.S.P.; Wu, A.T.; Shan, Y.S.; Wu, S.Y. Fucoidan inhibits radiation-induced pneumonitis and lung fibrosis by reducing inflammatory cytokine expression in lung tissues. Mar. Drugs 2018, 16, 392.
Reviewer 2 Report
Authors have shown that LMF can be used to down-regulate the expression of TGF-beta and related signal molecules up-regulated in the Radiation-induced fibrosis. There are two minor comments to improve quality of the manuscript.
Minor comments
1. Cell viability analysis was performed by MTT assay, but the viability of some samples (LMF) are more than 100%. In MTT assay, if the substance to be tested activates the cells, the results often include over 100%. Basically, cell viability is the ratio of living and dead cells, and more than 100% does not make sense. Since MTT reagent measures the activity of enzymes involved in cellular metabolic activity, it may be considered to be labeled as cellular metabolic activity instead of cell viability.
2. In Figure 5, the fluorescence intensity of A was quantified to make B and C graphs, but there is no method how to quantify it in Materials & Methods, and there is no C in legend.
Author Response
Reviewer 2
Comments and Suggestions for Authors
Authors have shown that LMF can be used to down-regulate the expression of TGF-beta and related signal molecules up-regulated in the Radiation-induced fibrosis. There are two minor comments to improve quality of the manuscript.
Minor comments
- Cell viability analysis was performed by MTT assay, but the viability of some samples (LMF) are more than 100%. In MTT assay, if the substance to be tested activates the cells, the results often include over 100%. Basically, cell viability is the ratio of living and dead cells, and more than 100% does not make sense. Since MTT reagent measures the activity of enzymes involved in cellular metabolic activity, it may be considered to be labeled as cellular metabolic activity instead of cell viability.
Response: Thank you
According to your comments, we have corrected as “cellular metabolic activity” rather than “cell viability” throughout the manuscript, Figure and figure legend with red mark.
- In Figure 5, the fluorescence intensity of A was quantified to make B and C graphs, but there is no method how to quantify it in Materials & Methods, and there is no C in legend.
Response: Thank you very much
The qualitative assessments of pSmad3 and Smad4 fluorescence intensity were used Image J software. And it was descripted in Line 380-382 to the manuscript. In addition, we also added (C) in Fig. 5 legend (Line 197) to the manuscript.
Reviewer 3 Report
Summary
This manuscript showed that low-molecular-weight fucoidan inhibited radiation-induced fibrosis in NIH3T3 cells through TGF-b1/Smad pathway. Low-molecular-weight fucoidan will be promising a treatment for radiotherapy. However, this manuscript should be improved and required for more data in order to be acceptable in Marine Drugs. My comments are written below.
Major comments
Could you show the stronger bands of PCR products in Fig.4A, and Fig.6A and B.? They are weak bands. If possible, you should analyze mRNA levels by using quantitative real-time PCR machine better. Because it is difficult to compare among bands using semi-quantitative PCR. Does irradiation induce Collagen I, Fibronectin, and a-SMA mRNA expression because I do not see increases of their mRNA expression in NIH3T3 cells after irradiation. You should re-think experimental condition, again.
Did you analyze p-Smad3 and Smad4 protein expression by western blotting? You should show p-Smad3 and Smad4 protein expression better. In addition, could you show clearer and brighter pictures in Fig.5A? It is not easy to see the pictures.
In figure7, you showed a significant increase of TGF-b1 and Collagen I protein expression in 1 Gy irradiation treated cells without LMF treatment compared with control without both irradiation and LMF treatment. I do not agree with you. I do not see increase of TGF-b1 protein expression in NIH3T3 cells after 1 Gy irradiation. Does TGF-b1 constitutively express in NIH3T3 cells? And it is a risky to quantify protein expression in comparison with between bands in this case. You should re-think experimental condition to increase TGF-b1 protein expression. In addition, did you examine effect of LMF treatment on TGF- b1 and Collagen I expression in NIH3T3 cells without irradiation? Because LMF suppressed TGF- b1 and Collagen I expression compared with control without irradiation and LMF treatment. Furthermore, is 25 mg/ml of LMF an optimal concentration in this experimental? This result does not agree to Fig.4A and Fig.6A.
After irradiation treatment, what type of factors increased TGF-b1 and Collagen I expression? p53, reactive oxygen species? Did you check them and examine effect of LMF treatment on them? I think it is important points in early phases after irradiation treatment.
Minor comments
I cannot find the figure legend about Fig.8C.
Make bar graphs carefully because the lines, showing statistically significant between columns, were shifted.
In Table 1, G3PDH should be changed to GAPDH.
Round 2
Reviewer 3 Report
I would like to accept the manuscript for publication in Marine Drugs. Thank you for your nice work.